# S-Adenosylmethionine Deficiency and Brain Accumulation of S-Adenosylhomocysteine in Thioacetamide-Induced Acute Liver Failure

**DOI:** 10.3390/nu12072135

**Published:** 2020-07-17

**Authors:** Anna Maria Czarnecka, Wojciech Hilgier, Magdalena Zielińska

**Affiliations:** Department of Neurotoxicology, Mossakowski Medical Research Centre, Polish Academy of Sciences, 5 Pawińskiego Street, 02-106 Warsaw, Poland; aczarnecka@imdik.pan.pl (A.M.C.); whilgier@imdik.pan.pl (W.H.)

**Keywords:** acute liver failure, cystathionine-β-synthase, glutathione, methionine adenosyltransferase 1A S-adenosylhomocysteine, S-adenosylmethionine, thioacetamide

## Abstract

Background: Acute liver failure (ALF) impairs cerebral function and induces hepatic encephalopathy (HE) due to the accumulation of neurotoxic and neuroactive substances in the brain. Cerebral oxidative stress (OS), under control of the glutathione-based defense system, contributes to the HE pathogenesis. Glutathione synthesis is regulated by cysteine synthesized from homocysteine via the transsulfuration pathway present in the brain. The transsulfuration-transmethylation interdependence is controlled by a methyl group donor, S-adenosylmethionine (AdoMet) conversion to S-adenosylhomocysteine (AdoHcy), whose removal by subsequent hydrolysis to homocysteine counteract AdoHcy accumulation-induced OS and excitotoxicity. Methods: Rats received three consecutive intraperitoneal injections of thioacetamide (TAA) at 24 h intervals. We measured AdoMet and AdoHcy concentrations by HPLC-FD, glutathione (GSH/GSSG) ratio (Quantification kit). Results: AdoMet/AdoHcy ratio was reduced in the brain but not in the liver. The total glutathione level and GSH/GSSG ratio, decreased in TAA rats, were restored by AdoMet treatment. Conclusion: Data indicate that disturbance of redox homeostasis caused by AdoHcy in the TAA rat brain may represent a deleterious mechanism of brain damage in HE. The correction of the GSH/GSSG ratio following AdoMet administration indicates its therapeutic value in maintaining cellular redox potential in the cerebral cortex of ALF rats.

## 1. Introduction

Oxidative stress and reduced glutathione play an essential role in the pathogenesis of several neurological disorders, including hepatic encephalopathy (HE). The brain, representing ~2% of body mass, consumes an extremely large fraction of the total oxygen (~20%) and therefore may be more vulnerable to oxidative damage than other organs [1]. Glutathione, a major antioxidant is synthesized as the final product in the transsulfuration pathway. This route converts homocysteine to cysteine and constitutes the metabolic link between antioxidant and methylation metabolism. The first and committing step in this pathway is catalyzed by cystathionine β-synthase (CBS), which is subject to complex regulation, including allosteric activation by the methyl donor, S-adenosylmethionine (AdoMet). The balance between preserving methionine via transmethylation under conditions of methionine restriction and committing it to transsulfuration under conditions of plenty is regulated at two key control points, methionine adenosyltransferase (MAT) and CBS [2].

AdoMet is synthesized from methionine and adenosine triphosphate (ATP) during the reaction catalyzed by one of three isoforms of enzyme AdoMet synthetase/MAT (MATI, MATII and MATIII). AdoMet functions as a principal donor of methyl groups in >100 different reactions catalyzed by methyltransferase enzymes [3,4] and participates in the following three types of reactions: transmethylation, transsulfuration and aminopropylation [5,6]. AdoMet supplementation restores the hepatic glutathione (GSH) pool and attenuates liver injury [5,7]. Eighty-five percent of all transmethylation reactions and more than 48% of methionine metabolism occur in the liver [8]. As a consequence, an injury of this organ could reduce the bioavailability of AdoMet and its consequent pathological effects.

During methyl transfer, AdoMet is converted to S-adenosylhomocysteine (AdoHcy), which can subsequently be hydrolyzed to form adenosine and homocysteine through a reaction catalyzed by AdoHcy hydrolase [9] occurring only after rapid product removal [10,11]. AdoHcy acts as a competitive inhibitor of AdoMet-dependent trans-methylation reactions, and its affinity to most methyltransferases is greater than the affinity of the substrate AdoMet. Therefore, AdoHcy removal is essential. Even though thermodynamics favoring the biosynthesis of AdoHcy, AdoHcy remains low in vivo provided that homocysteine is removed immediately [10]. The ratio of AdoMet to AdoHcy is predictive of cellular methylation status [12]. The reference values for AdoMet, AdoHcy, and AdoMet/AdoHcy ratio analyzed in plasma samples of 30 healthy donors revealed that mean ± SD concentrations were 93.6 ± 19.5 nM for AdoMet (range, 58–147 nM) and 20.9 ± 7.4 nM for AdoHcy (range, 8–40 nM). The resulting mean AdoMet/AdoHcy ratio was 4.9 ± 1.7 (range, 1.6–9.5) [13]. Certainly, the determination of these values for brain tissue is far more challenging.

In our previous study, a direct infusion of methyl donor, AdoMet, to the cerebral cortex of thioacetamide (TAA)-treated rats paradoxically decreased the extracellular concentration of asymmetrically methylated arginine residue, ADMA [14]. A strong feedback inhibition observed exclusively in TAA rats suggests that processes of AdoHcy removal are impaired in acute liver failure (ALF). These observations prompted us to expand our research and analyze the brain methylation index and the AdoMet-glutathione interdependence (Figure 1). Whereas lowered hepatic MAT expression and a beneficial effect of AdoMet supplementation after TAA exposure were confirmed in a few reports [15,16,17], none of them have focused on the brain. Therefore, our study aimed to assess whether (i) TAA-induced ALF in rats affects AdoMet and AdoHcy levels in the cerebral cortex and liver, (ii) exogenous administration of AdoMet is capable of modulating brain redox status expressed as a ratio of reduced to oxidized glutathione (GSH/GSSG).

## 2. Materials and Methods

### 2.1. Animals

Male Sprague Dawley rats of the initial body weight 250–300 g, from the outbred animal colony were supplied by the Animal House of Mossakowski Medical Research Centre, Warsaw, Poland. Animals were kept 2–3 in cages at room temperature (22 °C) under an artificial light/dark cycle (12/12 h), with access ad libitum to standard laboratory chow food and tap water. All procedures were carried out following directive 2010/63/EU and received prior approval no. 166/2016 from the 1st Local Ethics Committee for Animal Experimentation, Warsaw, Poland, as compliant with Polish Law (of 21 January 2005). All efforts were made to reduce the number of animals and to minimize their suffering. The study complies with the ARRIVE guidelines for reporting animal research.

### 2.2. Chemicals

Thioacetamide, S-adenosylmethionine, S-adenosylhomocysteine were provided by the Sigma-Aldrich Chemical Co. (Steinheim, Germany), S-(5-adenosyl)–L-methionine disulfate tosylate by Abcam (Cambridge, UK).

### 2.3. The Acute Liver Failure Model and S-Adenosylmethionine Administration

Animals received three intraperitoneal (ip) injections of TAA (300 mg/kg of body weight) per day for three consecutive days at 24 h intervals. Control rats were injected ip with saline. AdoMet (10 mg/kg, calculated as the free base) was administered three times subcutaneously immediately before TAA. The dose was selected based on literature reports [16,18]. Control rats were administered i.p. with saline analogically. Rats were decapitated 24 h after the third injection of TAA and AdoMet, and liver, whole cerebral cortex, and blood samples from all groups of animals were collected.

### 2.4. Determination of the Tissue Level of AdoMet and AdoHcy by HPLC-FD

The AdoMet and AdoHcy concentrations in the brain cerebral cortex and liver tissue were analyzed using HPLC with fluorescence detection. Tissue samples were weighed, homogenized in 9 volumes of ice-cold acidified methanol, and centrifuged (20,000× *g*, 20 min at 4 °C). The supernatants were collected and mixed with perchloric acid with chloroacetaldehyde and sodium acetate in a timed reaction (120 min, 60 °C), to obtain the corresponding 1,N6-etheno derivatives, as described earlier [19], with some modifications. Samples (60 µL) were injected onto 250 × 4.6 mm 5 µm Hypersil ODS column and eluted with a mobile phase of 0.1 M sodium acetate buffer, pH 4.5, containing 4.2% acetonitrile (v/v). The fluorescent AdoMet and AdoHcy 1,N6-etheno derivatives were monitored at excitation and an emission wavelength of 270 and 410 nm, respectively, and quantified by peak area comparisons with standard, run on the day of analysis. Fluorescence derivatives were separated by isocratic elution with a flow rate of 1 mL/min during 10 min, and then, the column was washed over 6 min by a linear increase of the acetonitrile content of the mobile phase from 4.5% to 50% during 1 min then within 1 min returned to the initial conditions. Data were collected and analyzed using a Chromeleon 6.8 software (Dionex, Sunnyvale, CA, USA).

### 2.5. GSSG/GSH Determination

Oxidized and reduced glutathione were measured using the Quantification kit for oxidized and reduced glutathione (38185, Sigma-Aldrich, Darmstadt, Germany) according to the manufacturer’s protocol. Briefly, 50 mg of tissue was homogenized in 0.2 mL of 5% SSA and centrifuged at 8000× *g*, 10 min, 4 °C. The supernatant was transferred to a new tube, and SSA concentration was reduced to 0.5% by the addition of ddH2O. Samples (40 µL) and GSH and GSSG standards were added into a microplate. For the determination of GSSG, only the “masking reagent” binding GSH was added to part of the samples. After 1 h incubation with buffer solution at 37 °C, the 20 µL of coenzyme solution and enzyme solution was added to each well. GSSG and total glutathione (GSH + GSSG) were detected spectrophotometrically at 415 nm. The GSH was quantified using the following equation: GSH = total glutathione − GSSG × 2.

### 2.6. Immunoblotting of CBS and MAT1A Proteins

Approximately 50 mg of rat brain cortex or liver tissue was homogenized in 10 volumes of RIPA buffer at 4 °C (150 mM NaCl, 1% IGEPAL, 0.5% sodium deoxycholate, 0.1% SDS, 50 mM Tris, pH 8.0) containing 50 mM sodium fluoride, Protease inhibitor cocktail (Sigma-Aldrich, St. Louis, MO, USA), and Phosphatase inhibitor cocktail (Sigma-Aldrich, St. Louis, MO, USA). The tissue homogenates were centrifuged for 10 min at 12,000× *g*, 4 C. Protein concentration in the supernatants was determined using the bicinchoninic acid protein assay kit (23227, Pierce Thermo Scientific, Rockford, IL, USA). Afterward, the denaturated samples containing 15 µg of total protein in 5 µL were fractionated by 10% sodium dodecyl sulfate-polyacrylamide gel electrophoresis (SDS–PAGE). Proteins from resolved gels were then transferred to nitrocellulose membranes. Blots were blocked with 3% bovine albumin serum in Tris-buffered saline Tween-20 buffer for 2 h. Incubation with CBS antibody (1:2000, Novus Biologicals, Centennial, CO, USA) or MAT1A antibody (1:1000, Novus Biologicals, USA) was done in Tris-buffered saline Tween-20 buffer with 1% bovine albumin serum at room temperature overnight at 4 °C followed by 2 h incubation with peroxidase-conjugated anti-mouse antibody (1:5000, Thermo Fisher Scientific, Rockford, IL, USA) or anti-rabbit antibody (1:8000, Sigma-Aldrich, St. Louis, MO, USA) for detection by Clarity™ Western ECL Substrate (Bio-Rad, Hercules, CA, USA). The first antibody was stripped off with 0.1 M glycine and pH 2.9, and the second incubation (1.5 h, 20–22 °C) was performed with an anti-GAPDH antibody (1:10000, Sigma-Aldrich, Aldrich, St. Louis, MO, USA). The amounts of protein per lane, as well as antibody concentrations, were optimized in pilot studies so that at least threefold differences in protein content were linearly reflected on immunoblots. The signals were visualized and quantified by the densitometric analysis with the FUJI-LAS 1000 system and Multi Gauge v. 3.0 software (Fujifilm, Tokyo, Japan). Results are presented as a percentage of the control of the analyzed protein: GAPDH ratio ± S.E.M.

### 2.7. Statistical Analysis

Biochemical parameters were analyzed by an unpaired Student’s t-test or a repeated-measures ANOVA with a post hoc Newman–Keuls test. The AdoMet/AdoHcy ratio was calculated for each individual rat. Values were expressed as mean ± SEM or as a% of the control group. A significance level of *P* < 0.05 was considered to be statistically significant. All statistical analyses were performed using Statistica for Windows v. 8.0 (Statsoft. Inc., Tulsa, OK, USA).

## 3. Results

### 3.1. Cerebral and Hepatic AdoMet and AdoHcy Concentrations

#### 3.1.1. Cerebral Cortex

HPLC analysis revealed a decreased tissue level of AdoMet by 21.97% (*p* < 0.001) accompanied by the increase in AdoHcy by 24.29% (*p* < 0.001) in the cerebral cortex of TAA rats (Figure 2a,b). The AdoMet/AdoHcy ratio, also known as methylation potential, was reduced in the TAA group by 35.34% (*p* < 0.001, Figure 2c).

#### 3.1.2. Liver

HPLC analysis revealed a decreased hepatic level of AdoMet (by 30.09%) and AdoHcy (by 21.53%) in TAA rats (Figure 3a,b). There was no change in the AdoMet/AdoHcy ratio between groups (Figure 3c).

### 3.2. CBS Protein Expression in the Rat Cerebral Cortex

The CBS protein level decreased by 43.11% (*p* < 0.05) in the brain cortex of TAA rats (Figure 4).

### 3.3. CBS and MAT1A Protein Expression in the Rat Liver

The CBS and MAT1A protein levels decreased by ~65.5%, *p* < 0.001 (Figure 5) and 40.45%, *p* < 0.05 (Figure 6), respectively in the liver of TAA rats.

### 3.4. The Effect of AdoMet (10 mg/kg sc) Treatment on the Total Glutathione Level and GSH/GSSG Ratio in the Cerebral Cortex

The total glutathione level decreased by 28.21% (*p* < 0.01) in TAA rats (Figure 7a) was restored by AdoMet treatment. GSH/GSSG ratio decreased by 28.34% (*p* < 0.01) in TAA rats was restored by AdoMet treatment (Figure 7b). There were no differences in the total glutathione level and GSH/GSSG ratio between control vs. control + AdoMet rats, and control + AdoMet vs. TAA + AdoMet rats.

## 4. Discussion

The main finding of this study pertains to the demonstration for the first time that AdoMet/AdoHcy balance is differently affected in the brain and liver by TAA-induced ALF in the rat. Furthermore, we expanded earlier reports from the peripheral organs to show that AdoMet treatment improves brain glutathione levels.

Whereas the AdoMet pool was reduced both in the brain and liver, AdoHcy accumulated only in the brain. The significant decrease in AdoMet led to the reduced AdoMet/AdoHcy ratio, known as the methylation index, a predictive value of reduced methylation capacity associated with an increase in AdoHcy. The potential maliciousness of AdoHcy lies in its high-affinity binding to the catalytic region of most AdoMet-dependent methyltransferases, which enables it to act as a potent product inhibitor. AdoHcy is converted to homocysteine and adenosine in a reversible reaction catalyzed by AdoHcy hydrolase that occurs only after rapid product removal [10,11]. It was documented that AdoHcy accumulation causes oxidative stress and neuronal excitotoxicity in neurodegenerative diseases [20]. Of note, AdoHcy-induced inhibition of methyltransferase enzymes is particularly important in the brain since the alternate homocysteine re-methylating enzyme, betaine-homocysteine methyltransferase, does not have any activity in the brain [20]. Therefore, it has been postulated, considering central nervous system (CNS) cells cannot export AdoHcy outside, that homocysteine extracellular transport is the only way CNS cells can decrease AdoHcy levels [20,21]. It is important to note that AdoHcy hydrolase levels are significantly lower in the brain compared to the liver or kidney [10,12]. The accumulation of AdoHcy observed exclusively in the brain of TAA rats suggests an impaired process of AdoHcy removal in the cerebral cortex that may likely be related to mechanisms described above.

Concomitantly, the observed decrease of CBS protein expression, the lowered total glutathione, and GSH/GSSG ratio in the cerebral cortex indicate the disturbances on the transsulfuration pathway. The direct cause is a decrease in the AdoMet level in the brain serving as the allosteric CBS regulator [2]. Under pathological conditions where AdoMet levels are diminished, CBS, and therefore glutathione levels, will be reduced. On the other hand, low AdoMet resulted from the decreased MAT1 in the liver where ~50% of the ingested methionine is metabolized [8]. Exposure to carbon tetrachloride [22] or lipopolysaccharide [23] has shown a relationship between redox stress-induced accumulation of inactive dimers or monomers of this enzyme [24] and MAT function. A dramatic decrease in liver-specific MAT activity and protein level in rats treated with TAA was previously documented [15,25].

The decreased AdoMet level and an impaired balance of AdoMet/AdoHcy indicates the importance of AdoMet supplementation. Although AdoMet is synthesized from methionine, the decreased MAT1 expression in the liver suggested that AdoMet but not methionine may be the best treatment choice. Indeed, elevated concentrations of circulating methionine in patients with liver disease have been reported [26], and excess methionine was shown to have toxic effects, including a decrease in hepatic ATP [5,27]. Numerous preclinical studies support a potential role for AdoMet in the treatment of chronic liver diseases [28] inter alia in reducing hepatic fibrosis [29,30] and in experimental liver injury [31].

A systematic review and meta-analysis support the efficacy and safety of AdoMet for the treatment of chronic liver diseases [7]. It has been suggested that AdoMet supplementation attenuates liver injury through the overregulation of GSH synthesis, reducing inflammation by the reduction of tumor necrosis factor-α and the regulation of the synthesis of interleukin-10 [32]. Moreover, a wide range of clinical studies provides a broad perspective on the role of AdoMet in the treatment of depression, neuropsychiatric disorders, and co-morbid medical conditions [33].

However, the direct effects of its supplementation on the brain are relatively poorly studied. Since AdoMet crosses the blood–brain barrier, with slow accumulation in the cerebrospinal fluid [34], the study of the central effects of AdoMet therapy in HE seemed reasonable.

In our study, the exogenous AdoMet normalized the glutathione level and reduced/oxidized form ratio in the cerebral cortex of TAA rats. In most tissues, transsulfuration pathway provides an important source of cysteine for GSH synthesis, but the brain is restricted by a low activity of cystathionine-γ-lyase (>100-fold lower than in the liver and kidney) [10] and a relatively high (~40-fold) cystathionine level compared to other human tissues [35]. Although it still contributes to GSH synthesis [1]. Interestingly, in the brain, the pathway for glutathione synthesis depends on metabolic integration between astrocytes and neurons and intersects with the glutamine–glutamate cycle [36]. The caveat is that our studies were limited to AdoMet/AdoHcy status in the cerebral cortex, and follow-up studies should evaluate other brain regions.

Regarding the CNS cell type, astrocytes are believed to be the major source of extracellular glutathione in the brain [37]. The neuronal dependence on the supply of astroglial cysteine might be important under oxidative stress conditions, in which the transsulfuration pathway is enhanced in astrocytes [1] and might represent the major source for neuronal cysteine needed to restore antioxidant capacity. Therefore, experimental evidence that exogenous AdoMet is capable of normalizing total GSH level and reduced/oxidized glutathione ratio in the brain is promising for alleviating the severity of HE.

## 5. Conclusions

TAA-induced ALF resulted in the accumulation of AdoHcy as well as a reduction of AdoMet and AdoMet/AdoHcy ratio in the brain. These led to the downregulation of CBS and a reduction of the GSH/GSSG pool. Pretreatment with AdoMet was shown to effectively preserve GSH levels. Since disturbance of redox homeostasis caused by AdoHcy in the TAA rat brain may represent a deleterious mechanism of brain damage in ALF, a detailed analysis of the methylation cycle in future studies to elucidate molecular mechanisms of AdoHcy impaired removal in the brain would be of interest. In general, the AdoMet treatment has been shown to restore brain oxidative status in rats with acute HE, suggesting that it could be used as a protective compound in brain injury triggered by severe liver failure.

## Figures and Tables

**Figure 1 nutrients-12-02135-f001:**
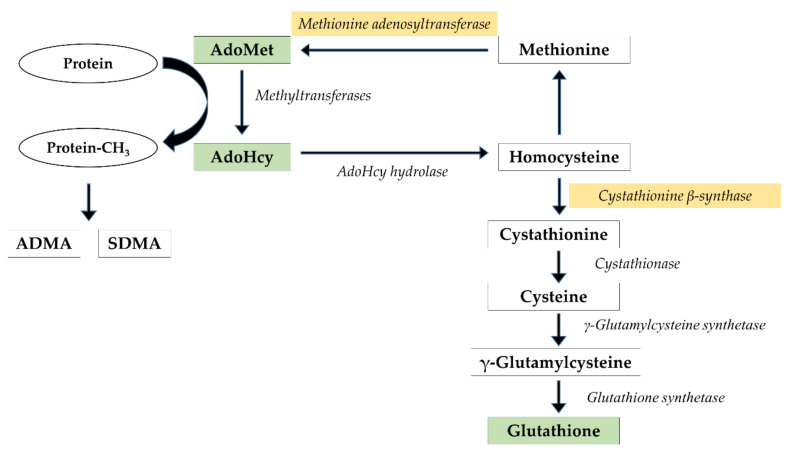
The methylation cycle and glutathione synthesis pathway. Metabolites and enzymes analyzed in the present study are shown in different colors.

**Figure 2 nutrients-12-02135-f002:**
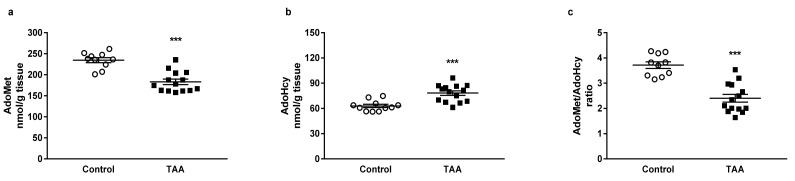
The AdoMet (**a**) and AdoHcy (**b**) tissue levels and (**c**) AdoMet/AdoHcy ratio in the cerebral cortex of control and TAA rats. All data are presented as an overlaying scatterplot of single data points ± SEM, *n* = 10–14. *** *p* < 0.001 compared with control.

**Figure 3 nutrients-12-02135-f003:**
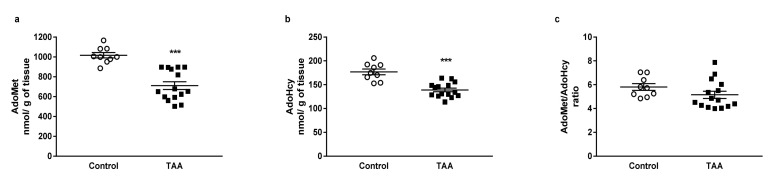
The AdoMet (**a**) and AdoHcy (**b**) tissue levels and AdoMet/AdoHcy (**c**) ratio in the liver of control and TAA rats. All data are presented as an overlaying scatterplot of single data points ± SEM, *n* = 9–15. *** *p* < 0.001 compared with control.

**Figure 4 nutrients-12-02135-f004:**
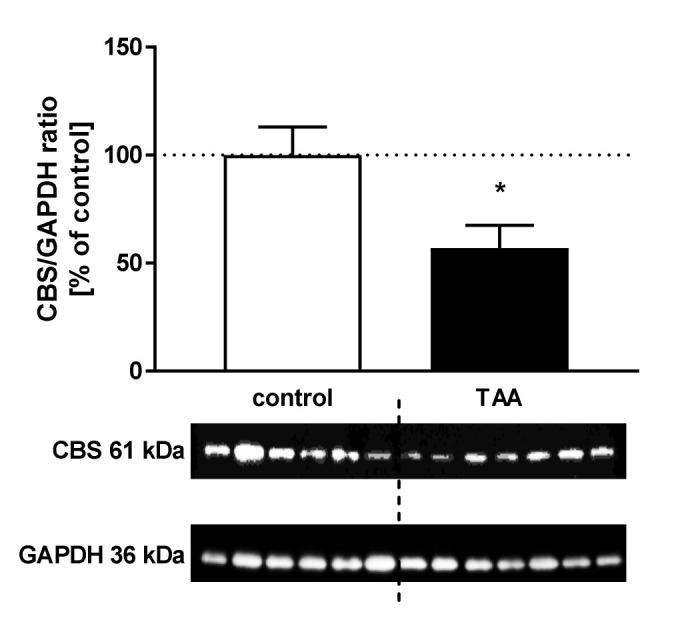
The relative protein level and representative immunoblot of cystathionine β-synthase (CBS) (61 kDa) in the cerebral cortex of control and TAA rats. The graph represents data normalized to GAPDH (36 kDa) and expressed as a percentage of control. Data are reported as mean ± SEM, *n* = 3–6. * *p* < 0.05 compared with control.

**Figure 5 nutrients-12-02135-f005:**
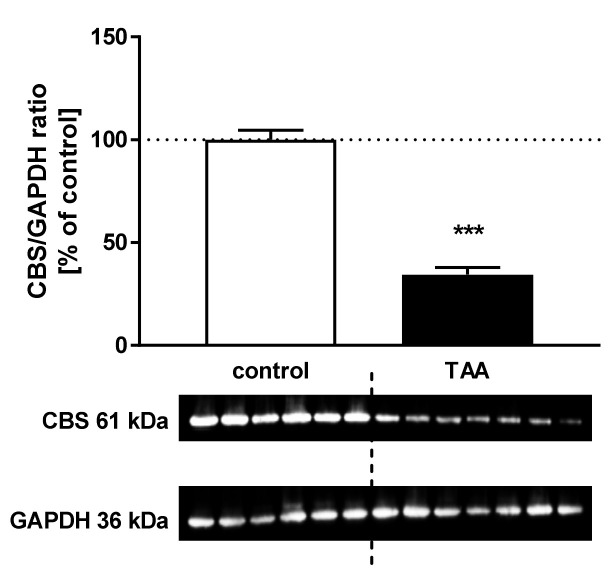
The relative protein level and representative immunoblot of CBS (61 kDa) in the liver of control and TAA rats. The graph represents data normalized to GAPDH (36 kDa) and expressed as a percentage of control. Data are reported as mean ± SEM, *n* = 6–7. *** *p* < 0.001 compared with control.

**Figure 6 nutrients-12-02135-f006:**
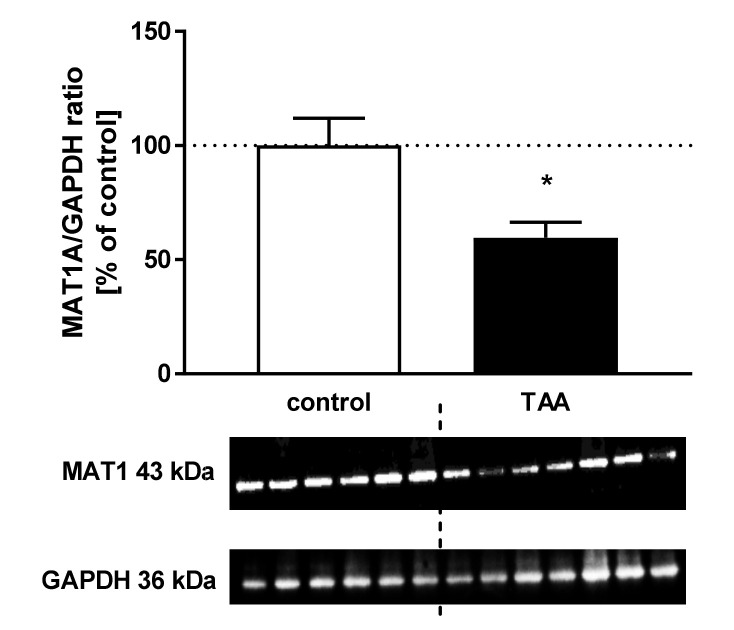
The relative protein level and representative immunoblot of MAT1 (43 kDa) in the liver of control and TAA rats. The graph represents data normalized to GAPDH (36 kDa) and expressed as a percentage of control. Data are reported as mean ± SEM, *n* = 6–7. * *p* < 0.05 compared with control.

**Figure 7 nutrients-12-02135-f007:**
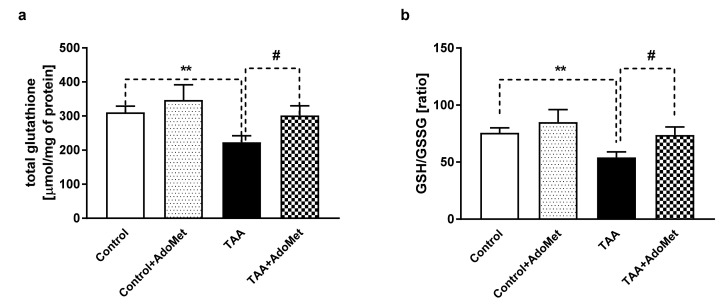
The effect of AdoMet (10 mg/kg sc) treatment on the total glutathione level (**a**) and glutathione (GSH/GSSG) ratio (**b**) in the cerebral cortex of rats with acute liver failure (ALF). Results are presented as the mean ± SEM, *n* = 4–9. ** *p* < 0.01, # *p* < 0.05 compared with control or TAA group, respectively.

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
