# Peer review of "S-Adenosylmethionine Deficiency and Brain Accumulation of S-Adenosylhomocysteine in Thioacetamide-Induced Acute Liver Failure"

_nutrients, 2020, doi:10.3390/nu12072135_

Round 1

Reviewer 1 Report

The manuscript by Anna et al., described in TAA induced acute liver failure rat, AdoMet, and AdoHcy concentration and ratio behaves differently in the brain than in the liver. The authors measured several additional biomarkers including CBS and MAT1 in homogenized tissue samples. The results are relatively interesting but lack resolution for future molecular mechanism studies. Detailed comments below:

  1. The general baseline AdoMet, AdoHcy, and their ratio at physiologically relevant normal conditions should be provided for reference.
  2. From our own previous experiences, GSH/GSSG kit is highly susceptible to oxidation during sample preparation. Proper control, especially recovery experiments should be added. For example, before measurement after tissue section, a known amount of GSH and/or GSSG should be mixed with the sample. Perform as the protocol described measurements, then compare samples with and without external GSH/GSSG. In this way, the authors should get a better understanding of how accurate the measurement is in their lab environment. We have observed a 20%+ error in some cases, especially GSSG due to its low concentration.
  3. How about contributions to GSH from other compensatory pathways? Cysteine can be imported from the bloodstream for GSH synthesis. When the deficiency is present, this may become active. Transporter activity could be an indicator of this. It has also been reported astrocytes can transport GSH to neurons. Since the bulk measurement did not separate glia with neurons, it is hard to conclude the reasoning behind the observation. An imaging or histology experiment should better answer this question.
  4. To add to the upper point, biochemistry characterization with the bulk tissue samples may mask many potential observations. An imaging experiment should be carried out to validate the results. Fixed sample staining of GSH, CBS, and other biomarkers used in the paper should be provided. Histology slices can be used to distinguish neurons from glia and visualize the differences. An even better experiment can include a live-cultured brain slice or acute slice with monitoring of these biomarkers during chemical treatments.
  5. What is the general suggestion in terms of application that can be deducted from the results? For example, how does this research benefit clinically?

Author Response

Response to Reviewer 1 Comments

We are grateful to Reviewer 1 for the critical review and for pointing out important suggestions and details on technical issues. We have now elaborated on these issues below.

Point 1: The general baseline AdoMet, AdoHcy, and their ratio at physiologically relevant normal conditions should be provided for reference.

Response 1: We thank the Reviewer 1 for this suggestion. Indeed this information is of importance in the context of our study. In the Introduction (p. 2; line 59-63) we provide the required details.

Point 2: From our own previous experiences, GSH/GSSG kit is highly susceptible to oxidation during sample preparation. Proper control, especially recovery experiments should be added. For example, before measurement after tissue section, a known amount of GSH and/or GSSG should be mixed with the sample. Perform as the protocol described measurements, then compare samples with and without external GSH/GSSG. In this way, the authors should get a better understanding of how accurate the measurement is in their lab environment. We have observed a 20%+ error in some cases, especially GSSG due to its low concentration.

Response 2: We thank the Reviewer 1 for this valuable suggestion. Indeed, the recovery experiments offer a better understanding of the accuracy of GSH/GSSG measurement. This recommendation will be of great importance in our further studies. In the present experiment, we analyzed the GSH/GSSG following the manual which did not suggest an additional recovery measurement. All samples were treated in the same way and a calibration curve for GSH and GSSG was determined in order to quantitative measurement of the effect of AdoMed treatment. Since GSH content in all samples was measured at the same time, accurate absolute values will not add to the significance of the results. Since, GSH/GSSG kit susceptibility to oxidation is relevant in the context of oxidation, as it was point by expert Reviewer 1, conducting of suggested validation in the same samples is unenforceable. This valuable comment will be of importance in the context of further experiments.

Point 3: How about contributions to GSH from other compensatory pathways? Cysteine can be imported from the bloodstream for GSH synthesis. When the deficiency is present, this may become active. Transporter activity could be an indicator of this. It has also been reported astrocytes can transport GSH to neurons. Since the bulk measurement did not separate glia with neurons, it is hard to conclude the reasoning behind the observation. An imaging or histology experiment should better answer this question.

Response 3: We thank the Reviewer 1 for this important comment. We are aware that the role of other compensatory pathways may be relevant as well, however detailed analysis of other contributing pathways was beyond the scope of this study. Indeed, L-cysteine is utilized in the synthesis of proteins, taurine, and GSH, and treatment of L-cysteine possesses among other, anti-fibrotic effects by inhibiting the activation and proliferation of hepatic stellate cells. In the CNS, L-cystine is transported across plasma membranes via two transporters: xc−, a cystine–glutamate exchanger, and XAG−, a family of high-affinity Na+-dependent glutamate transporters. For the synthesis of GSH, a metabolic interaction between neurons and astroglial cells is crucial. The study from our laboratory documented that L-cystine is transported to astrocytes almost exclusively by system XAG−, and ammonia, a major key neurotoxin in ALF, stimulated L-cystine uptake (DOI: 10.1016/j.neuint.2006.12.003) emphasizing the upregulation of L-cystine uptake as a factor contributing to astrocytic glutathione synthesis. The relative importance of L-cystine or L-cysteine as a precursor for GSH synthesis in neurons is however uncertain. We agree that visualization experiments including all analyzed biomarkers would be valuable and would rise the relevance of AdoMed treatment on GSH content in particular cell types of brain tissue. Of note in this context, the anti-GSH antibodies are a very useful tool for locating S-glutathionylated proteins in tissue sections (doi.org/10.1016/j.freeradbiomed.2017.08.008). The Reviewer’s suggestion would be implemented in our further study aiming to the analysis of S-glutathionylated proteins in specific cell types and intercellular distribution of GSH within them.

Point 4: To add to the upper point, biochemistry characterization with the bulk tissue samples may mask many potential observations. An imaging experiment should be carried out to validate the results. Fixed sample staining of GSH, CBS, and other biomarkers used in the paper should be provided. Histology slices can be used to distinguish neurons from glia and visualize the differences. An even better experiment can include a live-cultured brain slice or acute slice with monitoring of these biomarkers during chemical treatments.

Response 4: We agree with the Reviewer that more specific analysis aiming to distinguish cell types would be of great interest. This excellent suggestion, however, requires more detailed analysis (ex vivo or cultured brain slices preparation) which should be conducted within a frame of a separate project. The purpose of the present study didn’t include enzyme localization in specific cell types, therefore this type of experiment was not conducted. We are fully aware that cellular detection and localization of GSH would offer important knowledge regarding the differences in GSH levels in response to oxidative stress in subpopulations of cells. The usage of ThiolTracker™ Violet is planned to use in further experiments, since performing this type of examination in the present study is unenforceable. We documented the effectivity of AdoMed treatment in the context of GSH depletion observed in the cortex of ALF rats. Since astrocytes are the main target of ammonia toxicity, and CBS enzyme is localized predominantly in glial cells of the cerebral cortex (https://www.proteinatlas.org/ENSG00000160200-CBS/tissue) we presume that beneficial effect of AdoMed treatment may include predominantly glial cells and secondary, serving protection for neurons. This speculative statement will require further analysis.

Point 5: What is the general suggestion in terms of application that can be deducted from the results? For example, how does this research benefit clinically?

Response 5: The answer for this important question was not clearly stated indeed, now it is included in the Conclusion:In general, the AdoMet treatment has been shown to restoring brain oxidative status in rats with acute HE, suggesting that it could be used as a protective compound in brain injury triggered by severe liver failure”.

Reviewer 2 Report

In the manuscript ID: nutrients-848748 the effect of thioacetamide–induced acute liver failure on methylation cycle and gluthathione production was studied in the rat brain. The main finding of the study is the indication of different regulation of S-adenosylmethionine and S-adenosylhomocysteine in brain and liver tissue. Moreover, the evidence is provided that peripheral administration of S-adenosylmethionine improves brain glutathione level in thioacetamide–induced acute liver failure.  Both findings are original and novel.

Comments:

  1. It should be specified which part of cerebral cortex was investigated?
  2. Results, lines 162-164 should be deleted.
  3. Results: all data written as “ Ì´xx” (approximately) should be replaced by an exact value.
  4. Results, chapter 3.4.; the analyzed tissue should be specified.
  5. Fig. 2-6: the meaning of asterisks was not indicated.
  6. Fig. 2 b and Fig. 3 b; change vertical axis, fit scale to data values.
  7. Fig. 2 and Fig. 3: the title of figures is confusing; reedit it.
  8. Fig. 4-6: please explain that immunoblots are presented and describe the method of density measurement.
  9. Fig. 7: the statistical comparison between groups: TAA+AdoMet vs Control+AdoMet is lacking.
  10. I would suggest to replace term “Ado/Met supplementation” with: Ado/Met treatment or treated.

Author Response

We have revised our manuscript following the suggestions and comments made by Reviewer 2 in the following way:

Point 1: It should be specified which part of the cerebral cortex was investigated?

Response 1: We analyzed the whole cerebral cortex of each rat. This information is now provided in 2.3. section.

Point 2: Results, lines 162-164 should be deleted.

Response 2: We apologize for this overlooking. Lines 162-164 are deleted.

Point 3: Results: all data written as “ Ì´xx” (approximately) should be replaced by an exact value.

Response 3: In the current version of the manuscript we replaced all the data by an exact value.

Point 4: Results, chapter 3.4.; the analyzed tissue should be specified.

Response 4: The analyzed tissue is now specified through the Results section.

Point 5: Fig. 2-6: the meaning of asterisks was not indicated.

Response 5: We apologize for this overlooking. The meaning of asterisks is now indicated in the figure legend.

Point 6: Fig. 2 b and Fig. 3 b; change vertical axis, fit scale to data values.

Response 6: We thank the Reviewer 2 for this suggestion. Accordingly, we adjusted the vertical axis to the data values.

Point 7: Fig. 2 and Fig. 3: the title of figures is confusing; reedit it.

Response 7: Thank you for this suggestion. We rephrase the title of these figures.

Point 8: Fig. 4-6: please explain that immunoblots are presented and describe the method of density measurement.

Response 8: We rephrase the legends to the figures. The method of density measurement is included in 2.6. section.

Point 9: Fig. 7: the statistical comparison between groups: TAA+AdoMet vs Control+AdoMet is lacking.

Response 9: The statistical comparison between groups: TAA+AdoMet vs Control+AdoMet, was provided, and no significant changes between both groups were found.

Point 10: I would suggest to replace term “Ado/Met supplementation” with: Ado/Met treatment or treated.

Response 10: Accordingly we replaced the term “Ado/Met supplementation” with “Ado/Met treatment” in the text.

Reviewer 3 Report

The manuscript entitled “S-adenosylmethionine deficiency and brain accumulation of S-adenosylhomocysteine in thioacetamide-induced acute liver failure" by Anna Czarnecka , Wojciech Hilgier and Magdalena ZieliÅ„ska studied  redox homeostasis in hepatic encephalopathy condition.

The experimental model is very interesting and the pathway studied is equally intriguing. The study is well defined and experiments have been appropriately conducted.

Minor concerns are as follows:

A graphical abstract which describe the pathways before and after the AdoMet administration is highly recommended

The number of animals analysed for each experiment shall be included.

A detectable inflammation marker in the blood could be a good indicator to describe the effect of AdoMet administration

Author Response

We are happy to learn that expert Reviewer 3 acknowledges that our experimental model is very interesting and the pathway studied is equally intriguing.

 Point 1: A graphical abstract which describes the pathways before and after the AdoMet administration is highly recommended

Response 1: We provided a graphical abstract describing the effect of ALF on the brain before and after the AdoMed treatment.

 Point 2: The number of animals analysed for each experiment shall be included.

Response 2: In the current version of the manuscript we included the number of animals analyzed for each experiment.

Point 3: A detectable inflammation marker in the blood could be a good indicator to describe the effect of AdoMet administration

Response 3: We thank Reviewer 3 for valuable comments. We agree that a detectable inflammation marker in the blood could be a good indicator to describe the effect of AdoMet treatment. This valuable suggestion encouraged us to measure the level of IL-6 using flow cytometry in collected plasma samples frozen since previous experiments (Figure below). Unfortunately, over three-month sampling storage resulted in the measurement unreliable (large variations between samples) and control levels were below the detection limit, therefore additional, and also time-consuming experiments would have to be conducted to reliable document this effect.

IMAGE (Please see the attachment)

Fig. The effect of AdoMet treatment on the concentration of Il-6 in the plasma of rats with TAA-induced ALF.

Round 2

Reviewer 1 Report

The authors have answered all questions raised by reviewers. The manuscript can be published as it is.